# OpenReview forum: "Position: Public Health Systems Should Embrace a Multi-Layered Epidemic Early-Warning with LLM Agents and Local Knowledge Enhancement"
_ICML.cc/2025/Position_Paper_Track — Submitted to ICML 2025 Position Paper Track_

### Official Review · Reviewer_LrYz · 2025-03-13

**Significance:** 1
**Argument Clarity:** 3
**Rating:** 2
**Confidence:** 4

**Questions:**

None

**Discussion Potential:**

2

**Paper Summary:**

The paper proposes a multi-layered architecture for supporting early warning systems in an ethical manner. This includes integrating social media surveillance, local knowledge bases etc. They provide a systematic framework for thinking about the design of this problem (Table 1). The authors also demonstrate the potential of such a system by providing an actual instantiation of it and evaluating it on a few benchmark tasks.

**Position:**

Yes

**Position In Title:**

Yes

**Related Work:**

3

**Strengths And Weaknesses:**

Strength:
- The authors present a detailed three-tier architecture (front-end, mid-tier, back-end) with specific technical implementations for each layer. The integration of multiple ML techniques (LLMs, multi-agent systems, reinforcement learning, and federated learning) is well-integrated.
- The authors provide some empirical validation through a pilot study on COVID tweet classification, showing F1 score improvements from 0.89 (keyword filtering) to 0.97 with their RL-optimized approach.

Weakness:
My overall takeaway is that it is a good system design or public health position paper but I am not sure if it is a good fit for machine learning conference.

- The paper's core position is about restructuring healthcare infrastructure through a multi-layered surveillance system. While it incorporates machine learning technologies, the position has no relation with advancing ML methodology, algorithms, or theory or bringing a new perspective on the way it is practiced.
- While the paper briefly mentions "traditional epidemiological surveillance" as being "centered on clinical investigations, laboratory testing, and case reporting", it lacks a thorough analysis of existing epidemic early warning systems. The authors don't provide specific examples of current operational systems, their architectures, or their effectiveness metrics. In a position paper, it is expected that authors spend more time establishing their position in relation to what exists but it appears most of the emphasis is on their proposed architecture (not even the position).
- The paper doesn't adequately discuss how current public health decision-makers view early warning systems or what barriers have prevented the adoption of more advanced technological approaches, which is crucial context for their position.
- The paper lacks substantial discussion of implementation costs versus benefits, which would be crucial for convincing public health systems to adopt this approach.

**Support:**

2

---

> ### Author Rebuttal · Authors · 2025-03-31
>
> Dear Reviewer LrYz,
>
> We appreciate your insights in review. Here, we address the primary concerns you highlighted.
>
> ### 1. Suitability for a Machine Learning Conference
> We understand your comments on whether this paper is a strong fit for a machine learning (ML) -centric venue, given that we focus heavily on rethinking healthcare infrastructure rather than proposing a novel ML algorithm. Our intention, however, is to demonstrate how an existing array of ML techniques can be systematically integrated into a unified framework that addresses real-world public health requirements. Rather than contributing directly to a certain ML method, we aim to illuminate the interplay between these techniques and the broader epidemiological ecosystem. We see value in disseminating this perspective at an ML conference, as it showcases how established algorithms can drive domain-specific innovation when carefully orchestrated. While we do not claim fundamental ML breakthroughs, we believe that highlighting integration for high-stakes public health use cases can still spark constructive dialogue on potential research directions, including new forms of privacy-preserving computation or more reliable approaches to misinformation filtering.
>
> ### 2. Greater Analysis of Existing Early Warning Systems
> You noted that the paper briefly contrasts traditional surveillance methods—such as clinical and laboratory testing—but lacks detailed exploration of specific operational frameworks. We appreciate that this makes it harder for readers to evaluate where our proposal stands relative to real-world implementations. Although we do mention relevant research works and emphasize the need for improved timeliness and data diversity, we acknowledge that presenting concrete examples of widely used systems (for instance, the WHO’s Early Warning, Alert and Response System) would strengthen our position. We therefore plan to incorporate more substantial comparisons in on-going work, including the architectures and effectiveness metrics of major existing tools.
>
> ### 3. Perspectives of Public Health Decision-Makers and Barriers to Adoption
> We share your view that understanding the viewpoints of health authorities and the structural barriers they face is crucial to justifying any call for a revamped surveillance architecture. While we mention privacy concerns and data governance issues, our current discussion does not fully delve into the organizational factors (such as regulatory constraints, lack of domain expertise in AI, or limited digital infrastructure) that can impede the adoption of new technologies. In future iterations of this work, we intend to provide more systematic insights from interviews or secondary analyses on how health administrators perceive AI-assisted surveillance, as well as how policy, funding, and workforce training can directly affect uptake.
>
> ### 4. Cost-Benefit and Feasibility Considerations
> We agree that implementation costs, resource allocation, and clear-cut benefits are important elements for convincing real-world practitioners. Our paper currently outlines the potential advantages—faster detection, improved accuracy, enhanced privacy protection—but only briefly touches upon economic feasibility. To address this gap, we will add more specific discussion about operational overhead for each layer: front-end Agents typically require moderate computational resources, the mid-tier may require robust infrastructure for RL and vector databases, while the back-end demands expert labor and local knowledge base upkeep. Additionally, federated learning involves ongoing expenses, such as secure communication protocols and hardware-encrypted computing nodes, but these costs may be justified by the benefits of maintaining data sovereignty while still contributing to global outbreak intelligence.
>
> ### Summary
> We sincerely appreciate your recognition of our multi-layer design’s cohesiveness and the promise indicated by our pilot study. We also acknowledge your concerns regarding the paper’s relevance for an ML conference, its limited discussion of existing early warning systems, the importance of incorporating stakeholder perspectives, and the lack of an in-depth cost-benefit analysis. We believe these points can be effectively addressed through clearer comparisons to established tools, deeper engagement with public health stakeholders, and more concrete budgeting scenarios. Thank you for highlighting these areas, which will help us refine the paper into a stronger contribution for both public health and ML audiences. We trust these improvements will inspire a more favorable assessment of our manuscript.
> Again, we thank you for your insightful review comments.

---

### Official Review · Reviewer_nxCX · 2025-03-14

**Significance:** 2
**Argument Clarity:** 3
**Rating:** 2
**Confidence:** 5

**Questions:**

Please refer to "Strengths And Weaknesses"

**Discussion Potential:**

3

**Paper Summary:**

The paper's position is that public health systems should use their proposed three-layer framework for epidemic early warning. The framework consists of the following layers: (1)distributed multi-agent data ingestion; (2) centralized analytics; and (3) local knowledge base repository.

## update after rebuttal
After reading the authors' response and also other reviews, it seems like the concern of the method being overly complicated, and highly speculative is shared between the reviewers. I suggest that the paper focused on one or two aspect and take a clearer stance. I therefore, would like to keep my sore unchanged.

**Position:**

Yes

**Position In Title:**

Yes

**Related Work:**

3

**Strengths And Weaknesses:**

The paper states an interesting position and that is likely to stir the discussion. The paper states that the proposed framework is complementary to the existing approaches. The paper is generally well written and discusses background and related research sufficiently.

However, my main concern with this paper is that it discusses so many aspects and concerns, some of which are not even addresses by the proposed framework. For instance, it is unclear how the proposed method will address misinformation, which is one of the main challenges in this application. Also, even though the proposed method seems to be novel, the idea of using generative AI and multi agent systems for early warning has been well explored in the healthcare domain [1,2, 3,4], making the novelty of the proposed position limited. The paper needs to justify how the proposed position is different from the existing works.


[1]MacIntyre, Chandini Raina, et al. "Artificial intelligence in public health: the potential of epidemic early warning systems." Journal of International Medical Research 51.3 (2023): 03000605231159335.
[2]Tavana, Parisa, and Mohammad ZareiNejad. "Generative AI in Pandemic Prediction." Application of Generative AI in Healthcare Systems: 35.
[3]Tshilenge Mfumu, Jean-Claude, et al. "A Multiagent-Based Model for Epidemic Disease Monitoring in DR Congo." Biomedical Engineering Systems and Technologies: 11th International Joint Conference, BIOSTEC 2018, Funchal, Madeira, Portugal, January 19–21, 2018, Revised Selected Papers 11. Springer International Publishing, 2019.
[4]Castro, Brenno Moura, et al. "Multi-agent simulation model for the evaluation of COVID-19 transmission." Computers in Biology and Medicine 136 (2021): 104645.

**Support:**

3

---

> ### Author Rebuttal · Authors · 2025-03-31
>
> Dear Reviewer nxCX,
>
> We are grateful for your thorough review of our position paper. Below, we address the key points you raised under “Strengths and Weaknesses,” focusing on (1) how our framework deals with misinformation, and (2) how our position differs from existing works on using generative AI and multi-agent systems for epidemic early warning.
>
> ### 1. Responding to the Concern That Our Paper Covers Many Aspects, Some Not Fully Addressed
> We acknowledge that our paper touches on multiple challenges to demonstrate the broad scope of epidemic surveillance. Our intention is not to offer an all-encompassing, one-size-fits-all solution for each challenge but to underline how these problems interconnect and how our three-layer design can help mitigate several of them in an integrated pipeline. We are mindful that certain areas, especially misinformation handling, warrant more explicit discussion. However, we believe that addressing these issues in a holistic framework is precisely what distinguishes our proposed approach from narrower implementations that solve only isolated aspects of early-warning systems.
>
> ### 2. How the Proposed Framework Addresses Misinformation
> Misinformation in social media presents a major threat to any data-driven surveillance method, as it can lead to false positives, public panic, or overlooked genuine signals. While we do not claim to eliminate misinformation entirely, our architecture seeks to reduce its impact through iterative filtering and multi-agent cross-checking. In the front end, each agent collects raw inputs that are then semantically interpreted by LLMs. These LLMs, enhanced by consensus checks across agents, help flag likely hoaxes or contradictory claims—particularly if certain narratives fail to appear in multiple, independently verified streams. As the mid-tier leverages reinforcement learning (RL) to adjust similarity scores and detect spurious content, any systematic patterns of rumor propagation are penalized once the back-end experts identify false alarms. Over time, these negative rewards guide the RL agent to lower weights for data sources or content patterns prone to misinformation, making the pipeline adaptively robust. Though these processes may not completely immunize the system against sophisticated disinformation campaigns, they provide a structured approach to noise reduction, which standard data ingestion pipelines often lack.
>
> ### 3. Differentiation from Existing Generative AI and Multi-Agent Systems
> We appreciate your observation that using generative AI and multi-agent approaches for epidemic monitoring has precedents in the literature (e.g., [Your mentioned publications 1,2,3,4]). Our position is not that the notion of multi-agent or generative modeling is entirely new, but rather that our contribution lies in the synergy of three explicit layers (distributed ingestion, centralized vector-based RL refinement, and expert-driven local knowledge bases) to form a continual feedback loop. Prior works have primarily focused on either simulation-driven multi-agent modeling [Your mentioned publications 3,4] or generative AI for predictive tasks [Your mentioned publications 1,2]. By contrast, our framework inserts generative and cross-lingual LLMs into front-end data agents that collect and filter real-time signals (as opposed to retrospective modeling), applies RL to unify and refine these signals in MId-Tier, and then systematically harnesses local epidemiological validation in the Back-End. This multi-layered design explicitly tackles privacy constraints, human–AI collaboration, and region-specific nuance. We posit that this layered integration offers more practical advantages and thereby extends beyond standard generative or multi-agent approaches confined to narrower use cases.
>
> ### Summary
> We sincerely appreciate your thoughtful feedback. Thank you for acknowledging that our position is likely to spark discussion and that the paper is generally well written with sufficient background coverage. We also appreciate your reminder that issues such as misinformation warrant clearer explanation, and that our research’s novelty should be highlighted in relation to previous works. By clarifying how our three-layer model addresses misinformation through agent-based LLM filtering and RL-based adaptation, and by distinguishing our architecture from other generative or multi-agent systems, we aim to show how our proposal complements rather than replicates existing methods. We welcome continued dialogue to refine and extend our framework’s application to real-world public health challenges.
> Again, we thank you for your insightful review comments and trust these refinements will encourage a more favorable assessment of our manuscript.

---

### Official Review · Reviewer_chqi · 2025-03-24

**Significance:** 2
**Argument Clarity:** 3
**Rating:** 2
**Confidence:** 4

**Questions:**

1. Section 3.3 proposes multiple filtering processes, but how these filters interact with the LLM agents remains unclear. The agents seem primarily described as data crawlers rather than intelligent decision-makers. Can the authors explicitly clarify the value added by incorporating LLM agents, and justify their presence in the paper title? Is their functionality sufficiently critical compared to standard data collection methods?
2. Section 4.5 outlines metrics like spatiotemporal similarity and outbreak expansion rates. However, this discussion remains superficial. Could the authors elaborate more comprehensively on how the metrics are specifically chosen, quantified, and how can these metrics be integrated into the RL-based training and evaluation pipeline? Additionally, how would these metrics handle conflicting signals or noise from disparate sources?
3. The middle-layer design claims to output validated signals using RL optimization, yet the pilot study only experiment on sentiment classification. Given that epidemic signals are complex, can the authors clarify how stability and convergence in this RL-based system would be achieved, especially given multiple potentially conflicting optimization objectives? Has this approach been tested or simulated beyond sentiment analysis tasks?
4. The paper emphasizes human-in-the-loop validation, but practical workflows remain underspecified. Could the authors provide concrete examples or scenarios illustrating the integration between human expertise and automated LLM outputs? Specifically, how would workflows be structured, responsibilities delineated, and final decision-making authority managed?
5. How do the authors plan to address significant regulatory challenges inherent to such a globally distributed system, especially concerning data privacy laws, compliance across diverse jurisdictions, and varying levels of technological resources among public health institutions?

**Discussion Potential:**

2

**Paper Summary:**

This paper advocates for adopting a multi-layered epidemic early-warning system based on large language models, including multi-agent data ingestion, and locally enhanced knowledge bases. The authors propose a three-tier architecture: (i) distributed multi-agent data ingestion, (ii) centralized vector-based analytics and RL-driven optimization, and (iii) expert-validated local knowledge repositories. The paper argues this approach can accelerate epidemic detection, enhance accuracy, and safeguard privacy through federated learning and human oversight.

**Position:**

Yes

**Position In Title:**

Yes

**Related Work:**

3

**Strengths And Weaknesses:**

Strengths:
1. The topic is relevant and timely, aiming to address significant global public health challenges.
2. The paper acknowledges key concerns such as privacy, misinformation, and model reliability.
3. Some alternate views are discussed.

Weaknesses:
1. My primary concern is that the proposed design is overly high-level and speculative, which lacks concrete implementation pathways or feasibility analysis.
2. The proposed system integrates diverse data sources, sophisticated analytics, federated learning, and multi-layered human–machine interactions. Given this complexity, the pilot study conducted in the paper is not sufficient to justify the feasbiility of this ambitious design.
3. Despite emphasizing the necessity of local knowledge and human oversight, details on how local expertise would effectively interact with automated processes are insufficiently explored.

Minor typo: Section 4.3 and 4.4 are duplicated.

**Support:**

2

---

> ### Author Rebuttal · Authors · 2025-03-31
>
> Dear Reviewer chqi,
>
> We appreciate your thoughtful critique of our position paper. Below are condensed but focused responses to your five main questions.
>
> ### Q1: Filtering Processes in Section 3.3 and Relationship to LLM Agents
> A1: In our system design, each Agent goes beyond simple data crawling by applying an LLM-based semantic filter to interpret and categorize platform-specific content (text, images, video) before consensus checks. This cross-lingual, multimodal understanding reduces duplicates, flags misinformation, and adapts to new terminology—capabilities that purely keyword-based methods lack. LLM modules cluster semantically related posts, identify emerging disease expressions, and handle multiple languages and media. This ensures higher-quality candidate alerts for the Mid-tier and Back-End, confirming that LLM Agents offer advanced filtering rather than mere data scraping.
>
> ### Q2: Integrating Spatiotemporal Similarities and Outbreak Expansion Rates into RL, Especially When Signals Conflict
> A2: We embed each data point in a time-and-location vector space to detect clustering around symptoms and measure outbreak expansion rates by tracking how rapidly mentions grow. If content referencing similar symptoms converges around specific locations and time windows, this often indicates a possible outbreak. The expansion rate is computed by tracking how quickly relevant mentions or symptom clusters grow across short periods, offering a quantitative sense of outbreak velocity. These metrics feed into our RL-driven Mid-Tier, which treats the vector indexing and semantic matching processes as “Environment.” The RL agent dynamically tweaks thresholds for similarity scoring, weighting, and expansion detection. Data that turn out to be false positives incur negative reward signals once experts correct or dismiss them, which trains the system to adjust similarity thresholds and expansion-rate prioritization. In cases of conflicting reports, the RL algorithm learns, over successive feedback cycles, to lower confidence in inconsistent or short-lived spikes and to reinforce patterns consistently verified across multiple trusted sources.
>
> ### Q3: Stability and Convergence of the RL Approach Beyond Simple Classification
> A3: Although our pilot study (~80k social media posts) focused on COVID vs. non-COVID classification, the goal was to test RL-based threshold optimization with expert feedback against simpler filtering methods. Iterative expert corrections enhanced performance beyond static keyword or basic BERT setups. In more complex scenarios, the RL agent balances multiple indicators under multi-objective reward functions. Expert oversight and corrective loops help ensure convergence, preventing systematic bias or excessive false alarms.
>
> ### Q4: Practical Workflows for Human–AI Collaboration and Final Decision-Making
> A4: In our three-layered architecture, the Front-End Agents and Mid-Tier RL handle high-throughput tasks, but human experts remain decisive for final outbreak conclusions. When the RL engine flags high-risk signals, the Back-End houses expert teams perform deeper analysis. Validated alerts escalate to official warnings, while disproven cases penalize the RL agent, refining future judgments. By placing domain specialists at the final node, we prevent unjustified alarms, incorporate local context, and preserve trust in system outputs.
>
> ### Q5: Regulatory Challenges of a Global System, Including Privacy Compliance and Resource Disparities
> A5: We address multi-jurisdictional privacy by front-end anonymization and retaining patient-level data under local health institutions, conforming to regulations like GDPR. Federated learning (FL) or secure multi-party computation (SMPC) permits model updates across regions without centralizing sensitive data. Resource disparities are mitigated by lightweight front-end Agents for constrained environments, with deeper analytics handled by well-equipped centers. Local ethics committees guide permissible data exchange, ensuring legal compliance and public trust.
>
> ### Summary
> We sincerely value your feedback. By integrating LLM-driven multi-agents, RL strategies, and local knowledge bases, we aim to detect outbreaks earlier, enhance interpretive accuracy, and improve data governance. Your insights helped us refine workflow details and balance automation with expert oversight. We trust these clarifications address your concerns and illustrate our approach’s potential. Again, thank you for your invaluable comments, and we hope these refinements lead to a more favorable assessment of our manuscript.

---

### Official Review · Reviewer_ESoo · 2025-03-28

**Significance:** 2
**Argument Clarity:** 3
**Rating:** 3
**Confidence:** 4

**Questions:**

1. Can you provide more details on how the integration of federated learning and secure multi-party computation will address privacy and regulatory challenges across different regions?
2. Given the complexity and reliance on LLMs and reinforcement learning, what specific measures are in place to manage and mitigate the risk of false positives or misclassifications in high-stakes public health scenarios?

Minor suggestions:
1. ··expert modules” in line 200 with wrong double quotation marks
2. Replicated subsection 4.3.

**Discussion Potential:**

2

**Paper Summary:**

The paper advocates for a paradigm shift in epidemic surveillance by proposing a multi-layered early-warning framework that integrates large language model (LLM) agents with distributed multi-agent systems and locally maintained knowledge bases. Its core contributions include a three-tier architecture: a front-end that harvests and preliminarily screens multi-modal, multi-lingual data from diverse sources; a middle tier that refines these signals using vector-based semantic analysis and reinforcement learning (RL) for adaptive thresholding; and a back-end that incorporates expert validation via local knowledge bases while ensuring privacy through federated learning and secure multi-party computation. The authors argue that by coupling state-of-the-art AI techniques with domain expertise and privacy safeguards, public health systems can detect and respond to emerging outbreaks more swiftly and accurately, complementing traditional epidemiological methods.

**Position:**

Yes

**Position In Title:**

Yes

**Related Work:**

3

**Strengths And Weaknesses:**

Strengths of the paper include:
1. Its comprehensive and innovative integration of multiple technologies (LLMs, multi-agent systems, vector indexing, and RL) tackles the challenge of early epidemic detection.
2. The detailed architectural design provides a robust technical foundation.
3. The paper anticipates and addresses key concerns such as model reliability, privacy issues, and the necessity of augmenting rather than replacing existing methods, which enhances its relevance to the ICML community and encourages discussion on alternative views.

Weaknesses:
1. The complexity of the proposed system may pose significant challenges for real-world implementation, especially in resource-constrained settings.
2. The heavy reliance on emerging AI techniques like LLMs and RL, which can sometimes produce unpredictable outputs, could affect the system’s consistency in high-stakes public health scenarios.
3. Greater emphasis on scalability, detailed cost–benefit analysis, and strategies for integrating with existing public health infrastructures would further improve the work.

Suggestions for improvement include enhancing the discussion on operational challenges, expanding empirical evaluations, and providing clearer guidelines for regulatory compliance and integration into current systems.

**Support:**

3

---

> ### Author Rebuttal · Authors · 2025-03-31
>
> Dear Reviewer ESoo,
>
> Thank you for your thorough evaluation. Below, we response to your comments, particularly your questions regarding privacy-preserving methods (FL/SMPC) and safeguards against misclassifications.
>
> ### Q1: Federated Learning (FL) and Secure Multi-Party Computation (SMPC) Across Different Regions
> A1: We appreciate your interest in how our framework adheres to privacy regulations in a globally distributed context. In the Figure 3 (Back-End), we emphasize that patient-level information remains stored at local facilities. Rather than pooling raw data, each site trains partial model parameters (e.g., LLM embeddings or RL policy weights) and then encrypts or otherwise protects the resulting gradients before sending them to a central aggregator. This FL setup respects jurisdictional data sovereignty and the letter of regulations like the GDPR.
>
> To further reduce risks, we incorporate SMPC. Even if a malicious actor gained access to local updates, those updates are cryptographically split or encrypted. This ensures that neither the aggregator nor any peer node can inadvertently reassemble sensitive details about individual records. In essence, FL addresses “where” data is kept (locally) and “what” is shared (aggregate model parameters), while SMPC addresses “how” these parameters are kept private during transmission and aggregation. Combined, they allow multiple regions to continually refine a shared model without exposing personal data, thus meeting strict compliance demands in diverse legal environments.
>
> ### Q2: Mitigating False Positives and Misclassifications in High-Stakes Public Health
> A2: We acknowledge the critical need to control for unwarranted alarms in domains where public trust is paramount. Our updated figure highlights the Back-End as a distinct validation layer, staffed with epidemiologists and local domain experts. The proposed AI pipeline never issues final alerts autonomously, which handled by Human-AI Interaction mechanism with undergo expert review described in Figure 3.
>
> Moreover, the Mid-Tier’s RL-Driven Refinement continuously adjusts similarity metrics and detection thresholds based on real-time feedback from these experts. Whenever a misclassification (e.g., a rumor spike incorrectly flagged) is discovered, the RL mechanism registers a penalty, reducing the likelihood of repeating that mistake. Conversely, correct detections are rewarded. Over time, this loop significantly cuts down on false positives. In our pilot experiment with COVID vs. non-COVID tweet classification, we observed that RL-based threshold optimization performance better than normal keywords filtering method. While real outbreaks are more complex than simple classification tasks, the principle of continuous improvement via feedback remains central and scalable.
>
> ### Additional Improvements
> In response to your broader remarks on complexity and scalability, we plan to insert a dedicated subsection on Operational Challenges within the on-going work. This discussion covers:
> - Resource Constraints: Proposing lightweight agents for smaller clinics or underserved regions, where data is preprocessed minimally before advanced analytics are forwarded to well-resourced central nodes.
> - Institutional Readiness: Emphasizing the role of policy collaboration, funding channels, and workforce training to foster acceptance of the three-layer model in public health organizations.
> - Long-Term Maintenance: Stressing the importance of standardized data formats and iterative model upgrades to address emergent pathogens or rumor patterns.
>
> We also elaborate on potential synergy with existing epidemiological protocols (e.g., lab testing, syndromic surveillance), underlining that our architecture is intended to complement rather than supplant conventional practices. Precisely because false positives have broad social implications, the framework relies on domain experts for final sign-off.
>
> ### Summary
> We sincerely appreciate your insightful feedback and remain committed to addressing your concerns. By integrating multi-layered AI processing (LLM Agents and RL) with robust human expertise and privacy-preserving techniques (FL/SMPC), we aim to strike a practical balance between rapid outbreak detection and dependable, expert-verified decision-making. We are grateful for your comments, which prompted us to clarify our stance on data governance, expand our discussion of operational feasibility, and refine key architectural details. We look forward to further insights and continued exploration of how these methods can best serve real-world health needs. Again, we thank you for your invaluable review comments and hope these enhancements will inspire a more favorable assessment of our manuscript.

---

### Decision · Program_Chairs · 2025-04-27

**Decision:**

Reject

**Comment:**

The paper is a controversial among reviewers with the strengths and weaknesses summarized below. Reviewers like the paper as a detailed plan for an important application area but most the details are not too relevant to the broader ML community as represented by the reviewers. It also has questionable fit as a position paper.

Strengths
* The topic is relevant and timely, aiming to address significant global public health challenges
* comprehensive and innovative integration of multiple technologies
* paper anticipates and addresses key concerns such as model reliability, privacy issues, and the necessity of augmenting rather than replacing existing methods
* The authors provide some empirical validation through a pilot study on COVID tweet classification

Weakness
* The complexity of the proposed system may pose significant challenges
* The proposed design is overly high-level and speculative, which lacks concrete implementation
* discusses so many aspects and concerns, some of which are not even addresses by the proposed framework

My overall takeaway is that it is a good system design or public health paper but I am not sure if it is a good fit for machine learning conference.